# Sparse Gaussian Processes for Stochastic Differential Equations

**Prakhar Verma**
Aalto University
Espoo, Finland
prakhar.verma@aalto.fi

**Vincent Adam**
Aalto University
Espoo, Finland
vincent.adam@aalto.fi

**Arno Solin**
Aalto University
Espoo, Finland
arno.solin@aalto.fi

## Abstract

We frame the problem of learning stochastic differential equations (SDEs) from noisy observations as an inference problem and aim to maximize the marginal likelihood of the observations in a joint model of the latent paths and the noisy observations. As this problem is intractable, we derive an approximate (variational) inference algorithm and propose a novel parameterization of the approximate distribution over paths using a sparse Markovian Gaussian process. The approximation is efficient in storage and computation, allowing the usage of well-established optimizing algorithms such as natural gradient descent. We demonstrate the capability of the proposed method on the Ornstein–Uhlenbeck process.

## 1 Introduction

Dynamical systems in the real world are often well represented using stochastic differential equations (SDEs, [15]) incorporating the laws of physics and sources of stochasticity. They appear naturally in applications like finance, healthcare, gene modelling, *etc.* [4, 8]. An active area of research within the machine learning community is to develop algorithms to learn SDEs from observations of dynamical systems [2, 14, 6, 20]. Following these early works, we frame the SDE learning problem as an inference problem: maximizing the marginal likelihood of observations under a generative model of the unobserved path (SDE prior) and the observations. For non-linear SDEs, this problem is intractable, so we use the variational inference framework [3] to derive and approximate the posterior.

As in Archambeau et al. [2], we introduce an approximate posterior process over paths in the form of a multi-output Markovian Gaussian process and frame the inference and learning problem as the maximization of a lower bound to the marginal likelihood (ELBO). This particular choice of the approximate posterior process leads to a tractable ELBO that can be efficiently evaluated and optimized. Crucially, it exploits the fact that the marginal statistics (mean and covariance) of Markovian Gaussian processes are obtained in closed form and cheaply by solving simple linear ordinary differential equations (ODEs). In practice, the Markovian Gaussian process is parameterized as a time-varying linear SDE and discretized on a fine temporal grid, leading to further approximations, and high storage and computation costs.

In this work, we propose an alternative parameterization to the approximate distribution over paths using a *conditioned* stationary Markovian Gaussian process, inspired by the doubly-sparse Gaussian process [1]. The key idea is to learn *pseudo*-observations such that a simple stationary GP conditioned on these *pseudo*-observations provides a good approximation to the intractable posterior, as measured by the Kullback–Leibler (KL) divergence. The proposed approximation reduces the complexity both in memory and time, allowing the usage of well-established optimizing algorithms such as natural gradient descent. The capability of the proposed method is demonstrated on the Ornstein–Uhlenbeck (OU) process.

35th Conference on Neural Information Processing Systems (NeurIPS 2021).

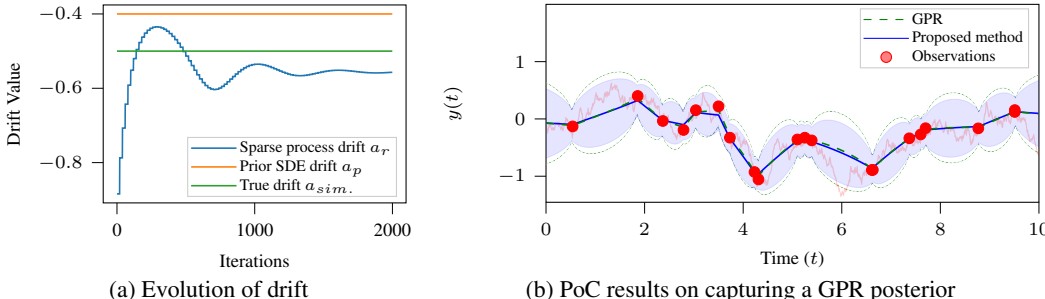

Figure 1: Ornstein–Uhlenbeck process: (a) The evolution of the drift of the sparse Markovian Gaussian process over iterations along with the prior SDE and the true SDE drift; (b) GPR posterior and approximated posterior mean and 95% confidence interval of the proposed method along with the simulated trajectory and the noisy observations.

This work is a direct extension of Archambeau et al. [2] which performs variational inference over the latent state path using a Gaussian Process as an approximate posterior process. Within the variational framework, alternative parameterizations for the posterior process have been used. In Li et al. [10], the drift of a non-linear SDE is parameterized. The resulting ELBO is not tractable but posterior sample path can be approximately generated (after a discretization of the time axis) to provide unbiased estimate of the ELBO and its gradient for stochastic optimization. Our approach bears similarity with the probabilistic numerics approach to solve or fit ODEs to data, whereby the solution is cast as an inference in a generative model with Markovian GP prior over the solution and two likelihoods: one enforcing a fit to observed data and a second enforcing 'gradient-matching', *i.e.* the gradient of the process is in agreement with the ODE [16, 18]. Both terms arise naturally in our framework in the form of the expected log-likelihood of the observations under the posterior process (variational expectations) and a distance between prior and posterior drifts (Girsanov term), respectively the first and second term in Eq. (2).

The contributions of this paper are: *(i)* We provide an alternate parameterization to the approximate distribution over paths using a *conditioned* Markovian Gaussian process. *(ii)* The proposed approximation leads to a more efficient method both in terms of memory and time. *(iii)* The proposed method catalyzes the usage of well-established and efficient optimizing algorithms such as natural gradient descent.

## 2  Methods

We model an observed dynamical system on a time interval $[0, \tau]$ using an SDE: $d\mathbf{x}_t = f_\theta(\mathbf{x}_t, t)\, dt + L\, d\beta_t$, where $f_\theta(\mathbf{x}_t, t)$ is the drift function, $LL^\top = \boldsymbol{\Sigma}$ is the (time-invariant) diffusion coefficient, and $d\beta_t$ is the standard Brownian motion. We focus on systems where the diffusion term is constant, and the state $\mathbf{x}$ is indirectly observed at $n$ discrete time points $t_i$ via an observation model providing the likelihood $\{p(\mathbf{y}_i \mid \mathbf{x}_i)\}_{i=t_1}^{t_n}$. The aim is to learn the $\theta$ parameter(s) of $f_\theta(\mathbf{x}_t, t)$ given observations by maximizing the marginal likelihood $p_\theta(\mathbf{y}_{t_1,\ldots,t_n})$. We consider the scenario where the model has arbitrary likelihood, and the drift of the SDE $f_\theta(\mathbf{x}_t, t)$ is non-linear. Computing the posterior distribution over state paths and the marginal likelihood is intractable, we thus resort to an approximate inference scheme: variational inference [3].

### 2.1  Variational inference

Variational inference (VI) turns an inference problem into an optimization problem. By introducing a distribution $q$ over paths, a lower bound to the log-evidence $\mathcal{L}(q) \leq \log p(\mathbf{y})$ is constructed via Jensen's inequality: $\mathcal{L}(q) = \mathbb{E}_q \log \frac{p(\mathbf{y}, \mathbf{x})}{q(\mathbf{x})} = \mathbb{E}_q \log p(\mathbf{y} \mid \mathbf{x}) - D_{KL}[q(\mathbf{x}) \| p(\mathbf{x})]$. This bound is optimized for $q \in \mathcal{Q}$, where $\mathcal{Q}$ is a family of distributions chosen to lead to a tractable bound. We will refer to this bound as the evidence lower bound (ELBO). The gap in the bound can be shown to be the KL divergence between the $q$ and the true posterior, $\log p(\mathbf{y}) - \mathcal{L}(q) = D_{KL}[q(\mathbf{x}) \| p(\mathbf{x} \mid \mathbf{y})]$. Thus, the optimal $q^* = \arg\min \mathcal{L}(q)$ also provides an approximation to the posterior $p(\mathbf{x} \mid \mathbf{y})$.

We choose the approximating distribution family $\mathcal{Q}$ to be that of Gaussian processes (GP, [13]). In this setting, valid Gaussian processes are Markovian and correspond to the class of linear SDEs. Archambeau et al. [2] proposed using markovian Gaussian process for $q$ by directly parameterizing the drift of the SDE as an affine function of the state: $q(\mathbf{x}(\cdot))$ : $\mathrm{d}\mathbf{x}_t = f_L(\mathbf{x}_t, t) + \sqrt{\mathbf{\Sigma}}\,\mathrm{d}\beta_t$, where $f_L(\mathbf{x}_t, t) = -A_t\,\mathbf{x}_t + b_t$, and $A_t, b_t$ are functions of time referred to as the variational parameters. Note that the diffusion term is set to the prior diffusion which is necessary to obtain a valid bound. In practice, optimizing over functions $A_t, b_t$ requires further assumptions or approximations. Archambeau et al. [2] resort to the later and discretize the continuous time SDEs, of both the prior and approximate posterior, over a fine time grid. This turns functions $A_t, b_t$ into vectors which can be optimized using standard optimization techniques, albeit at the expense of modifying the *prior* assumptions on the dynamical system.

We now propose an alternative parameterization for $q$ that does not require to approximate the prior SDE. We do so by choosing $q$ to be a *conditioned* Markovian GP (or sparse Markovian GP) built by conditioning the states $\mathbf{x}(z)$ of a stationary Markovian GP $r_\phi$ at time indices $z$ to a Gaussian variable with distribution $w_\psi$. We refer to $z$ as inducing inputs and $\mathbf{x}(z)$ as inducing states. Informally, this leads to a factorization of the density over paths, $q_{\{\phi, \psi\}}(\mathbf{x}(\cdot)) = r_\phi(\bar{\mathbf{x}}(\cdot) \mid \mathbf{x}(z))\,w_\psi(\mathbf{x}(z))$, where $\mathbf{x}(\cdot)$ are the states for all time inputs and $\bar{\mathbf{x}}(\cdot) = \mathbf{x}(\cdot) \setminus \mathbf{x}(z)$, *i.e.*, all states except the inducing states $\mathbf{x}(z)$ at inducing input $z$. The Markovian GP $r_\phi(z)$ can be represented as an LTI-SDE [15]; $r_\phi(\mathbf{x})$ : $\mathrm{d}\mathbf{x}_t = f_\phi\,\mathbf{x}_t\,\mathrm{d}t + L\,\mathrm{d}\beta_t$, and, as in Archambeau et al. [2] we restrict the diffusion term to be the same as that of the prior SDE. Thus, $\{\phi, \psi\}$ are the variational parameters.

The ELBO introduced in Section 2.1 requires the computation of the KL divergence between the approximate posterior and the true posterior processes. For Markovian processes, this can be done using Girsanov theorem [7],

$$\mathrm{D}_{\mathrm{KL}}\left[q(\mathbf{x}) \,\|\, p(\mathbf{x})\right] \;=\; \frac{1}{2}\int_{t=0}^{\tau}\mathbb{E}_{q(\mathbf{x}_t)}\|f_\theta(\mathbf{x}_t) - f_\phi\,\mathbf{x}_t\|_{\mathbf{\Sigma}^{-1}}^2\mathrm{d}t + \mathrm{D}_{\mathrm{KL}}\left[w(\mathbf{x}(z)) \,\|\, r(\mathbf{x}(z))\right]. \quad (1)$$

Thus, the ELBO for the proposed model is

$$\mathcal{L} = \sum_{i=0}^{n}\mathbb{E}_{q(\mathbf{x}(t_i))}[l(\mathbf{x}_i)] + \int_{t=0}^{\tau}\mathbb{E}_{q(\mathbf{x}_t)}\left[g(\mathbf{x}_t)\right]\mathrm{d}t - \mathrm{D}_{\mathrm{KL}}\left[w(\mathbf{x}(z)) \,\|\, r(\mathbf{x}(z))\right], \quad (2)$$

where $g(\mathbf{x}_t) = -\frac{1}{2}\left(f_\theta(\mathbf{x}_t) - f_\phi\,\mathbf{x}_t\right)^\top\mathbf{\Sigma}^{-1}\left(f_\theta(\mathbf{x}_t) - f_\phi\,\mathbf{x}_t\right)$, and $l(\mathbf{x}_i) = \log p(\mathbf{y}_i \mid \mathbf{x}_i)$, with the observations assumed independent and identically distributed. The ELBO in Eq. (2) can be further written as $\mathcal{L} = \mathcal{L}_{\mathrm{sde}} + \mathcal{L}_{\mathrm{svgp}}$, where $\mathcal{L}_{\mathrm{sde}} = \int_{t=0}^{\tau}\mathbb{E}_{q(\mathbf{x}_t)}\left[g(\mathbf{x}_t)\right]\mathrm{d}t$, and $\mathcal{L}_{\mathrm{svgp}} = -\mathrm{D}_{\mathrm{KL}}\left[w(\mathbf{x}(z)) \,\|\, r(\mathbf{x}(z))\right] + \sum_{i=0}^{n}\mathbb{E}_{q(\mathbf{x}(t_i))}[l(\mathbf{x}_i)]$ which is identical to the ELBO of the SVGP model [17], considering $r$ as the pseudo prior. The ELBO can be interpreted intuitively. It consists of two parts: $\mathcal{L}_{\mathrm{sde}}$ aims to keep the Markovian GP $r$ close to the original prior SDE, whereas $\mathcal{L}_{\mathrm{svgp}}$ aims to learn the variational parameters $\{\phi, \psi\}$ considering $r$ as the prior. A key feature of our approach is that marginal posterior predictions $q(x(t))$ necessary to evaluate the ELBO can be computed in parallel for all time inputs $t$, unlike in Archambeau et al. [2] where those statistics require classical sequential Kalman smoothing recursions.

## 2.2 Optimization

The ELBO is optimized in a two-step iterative algorithm, following the variational EM algorithm [11], as shown in Alg. 1. We use gradient descent to learn the $\theta$ parameters of the prior SDE whereas for inference, *i.e.* learning $q$, natural gradient descent is used for parameters $\psi$ of the distribution $w_\psi$ and gradient descent for $\phi$ parameters.

Natural gradient descent can be used when optimizing an objective over a distribution. The resulting optimization is invariant to the choice of parameterization. We use the formulation of natural gradient descent as mirror descent [12] in the *mean parameterization* which provides an update for the the *natural parameterization* of the distribution [9] (See App. A for a description of parameterizations of the multivariate normal distribution). Noting $\boldsymbol{\eta}_r$ the natural parameters of $r(\mathbf{x}(z))$ and parameterizing $w(\mathbf{x}(z))$ in the natural form $\boldsymbol{\eta} = \boldsymbol{\eta}_r + \boldsymbol{\lambda}$, we get the natural gradient updates:

$$\boldsymbol{\lambda}_{t+1} = \gamma_t \nabla_{\boldsymbol{\mu}}\,\alpha + (1 - \gamma_t)\,\boldsymbol{\lambda}_t, \quad (3)$$

where $\gamma_t = \frac{1}{1+\rho_t}$, and $\alpha = \int_{t=0}^{\tau}\left(\mathbb{E}_{q(\mathbf{x}_t)}\left[g(\mathbf{x}_t)\right] + \sum_{i=0}^{n}\delta(t - t_n)\,\mathbb{E}_{q(\mathbf{x}(t_i))}[l(\mathbf{x}_i)]\right)\mathrm{d}t$, with $\boldsymbol{\mu}$ being the mean parameter, $\boldsymbol{\lambda}$ the natural parameter of $w$, and $\delta$ is the dirac function. The gradient of $\alpha$ is available in closed-form via the chain-rule (see App. B). It takes the form of a time integral which we approximate via Riemann sum.

## 3 Experiments

We showcase the inference capability of our method on the Ornstein–Uhlenbeck process. The inducing variables are taken to be same as the observations points and are not optimized. Also, the learning step is not performed for this experiment. However, both of these can be easily integrated.

The Ornstein–Uhlenbeck (OU) process is a stochastic process of a particle going through a Brownian motion [19]. It is defined by a stationary Markovian GP expressed by an SDE, $d\mathbf{x}(t) = -a\,\mathbf{x}(t)\,dt + \sigma\,d\beta(t)$, where drift function is $f(\mathbf{x}_t) = -a\,\mathbf{x}_t$, diffusion function is $\sigma$, and Brownian motion has $q$ spectral density. We simulate the OU SDE using Euler–Maruyama and observe states at random time-intervals via a Gaussian likelihood observation model. For the experiment, likelihood variance is fixed and not optimized. More details about the experiment setup are given in App. C. The induced stationary covariance function of OU process is $\kappa(t,\,t') = \frac{\varphi}{2\lambda}\exp(\lambda\|t - t'\|)$, where $\varphi = \sigma^2\,q$, which is identical to the Matérn 1/2 kernel. Thus, we perform Gaussian process regression (GPR) with Matérn-½ kernel to get the exact posterior. We apply the proposed method to approximate the posterior with $q(\mathbf{x}(\cdot)) = r(\mathbf{x}(\cdot) \mid \mathbf{x}(z))\,w(\mathbf{x}(z))$, where the kernel of $r$ is the modified Matérn-½; whose diffusion coefficient matches that of the prior SDE.

---

**Algorithm 1:** Optimization

$\eta,\ \nu,\ \gamma \leftarrow$ learning rates
**while** *not converged* **do**
    $\theta_{n+1} \leftarrow \theta_n + \nu\,\nabla_\theta\,\mathcal{L}_{\mathrm{sde}}$
    **while** *not converged* **do**
        **while** *not converged* **do**
            *Natural gradient step:*
            $\bar{\lambda}_{n+1} \leftarrow \gamma_t\nabla_{\boldsymbol{\mu}}\,\alpha + (1 - \gamma_t)\,\bar{\lambda}_n$
        **end**
        *Hyperparameter gradient step:*
        $\phi_{n+1} \leftarrow \phi_n + \eta\,\nabla_\phi\,\mathcal{L}$
    **end**
**end**

---

App. C(Fig. 3) showcases the evolution of the ELBO over iterations during optimization as well as of its different components. The evolution of the drift of the sparse Markovian Gaussian process $r$ is shown in Fig. 1a from which we infer that the drift converges to a good approximation. The posterior approximated by the proposed method along with the GPR posterior is shown in Fig. 1b which showcases the capability of the model to approximate the posterior which is very close to the exact GPR posterior.

## 4 Limitations and Extensions

The proposed method can be summarized as performing GP regression with a *pseudo* Markovian GP prior, while ensuring that the drift of this *pseudo* prior matches that of the prior SDE. A stationary GP has a linear drift and can not be expected to approximate well a non-linear drift. For example, as is, the proposed method could not learn the double well system of Archambeau et al. [2] whose drift is saw-tooth like. A natural extension, which we are currently investigating, is to use a piecewise stationary Markovian GP whose drift coefficient is different in between each consecutive pair of inducing points $f_{\phi,t} = \sum_m \delta(z_m < t \le z_{m+1})f_{\phi_m}$. Each sub-drift $f_{\phi_m}$ would thus only approximate the prior SDE drift locally in time, and thus locally in the state-space. Alternatively, a mixture of Markovian GPs could be used which would, once learned, automatically cluster the state-space to provide a global approximation to the prior drift as in Fox et al. [5].

## 5 Conclusion

In this paper, we proposed a method to learn the SDE based on a set of noisy observations. The focus was on non-linear SDEs with a complex observation model leading to an intractable posterior. Gaussian processes (GPs) are often used as approximate posterior over SDE paths. However, the resulting algorithm has a high number of parameters with high complexity both in terms of storage and time. We explore the advances related to sparse GPs and present a novel alternate parameterization to the approximate distribution over SDE paths based on a sparse Markovian Gaussian process. The proposed method has fewer parameters, the ELBO calculation is parallelizable in contrast to the current methods, and allows the usage of well-defined optimization algorithms such as natural gradient descent for better convergence. We demonstrated the model capability on an Ornstein–Uhlenbeck process, for which the 'ground-truth' is available. The results show that the new approach works as intended and encourages further research on the applicability of this method.

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

# Supplementary Material:
# Sparse Gaussian processes for stochastic differential equations

## A  Multivariate normal parameterizations

The multivariate normal (MVN) is often parameterized in terms of its *source* parameters: the mean and the covariance matrix $(\mathbf{m}, \mathbf{S})$. The MVN distribution is part of the exponential family which provides additional parameterizations of interest. Distributions in the exponential family have densities of the form

$$p(\mathbf{x}) = \exp(t(\mathbf{x})^\top \boldsymbol{\eta} - a(\boldsymbol{\eta})), \tag{4}$$

where $t(\mathbf{x})$ are the sufficient statistics, $\boldsymbol{\eta} \in \mathbb{R}^d$ the natural parameters, and $a(\boldsymbol{\eta})$ the log partition function defined by $a(\boldsymbol{\eta}) = \log \int \exp(t(\mathbf{x})^\top \boldsymbol{\eta})) d\mathbf{x}$. For a given natural parameterization $\boldsymbol{\eta}$, there is an associated expectation parameterization $\boldsymbol{\mu} = \mathbb{E}_{\boldsymbol{\eta}}[t(\mathbf{x})]$. For the MVN distribution, the sufficent statistics are given by $t(\mathbf{x}) = (\mathbf{x}, \mathbf{x}\mathbf{x}^\top)$ and the natural parameters in terms of source parameters are $\boldsymbol{\eta} = (\mathbf{S}^{-1}\mathbf{m}, -1/2\mathbf{S}^{-1})$.

## B  Method

### B.1  Variational posterior and chain rule

Similar to Adam et al. [1], using the state-space parameters, the conditional of the sparse Markovian GP is $r(\mathbf{x}_t \mid \mathbf{x}_z) \sim \mathcal{N}(\mathbf{P}_t\, v_t, \mathbf{T}_t)$, where $v_t = (u_{t-}, u_{t+})$ are the inducing variable pairs, and $\mathbf{P}_t$ and $\mathbf{T}_t$ are the matrices depending on the previous state transitions. With the probability density of the inducing variables being Gaussian, $w(\mathbf{x}_z) \sim \mathcal{N}(\boldsymbol{\mu}_{w_z}, \boldsymbol{\Sigma}_{w_z w_z})$, the variational posterior is $q(\mathbf{x}_t) \sim \mathcal{N}(\boldsymbol{\mu}_t, \boldsymbol{\Sigma}_t)$, where $\boldsymbol{\mu}_t = \mathbf{P}_t\, \boldsymbol{\mu}_{w_t}$ and $\boldsymbol{\Sigma}_t = \mathbf{T}_t + \mathbf{P}_t \Sigma_{w_t w_t} \mathbf{P}_t^\top$.

Using the variational posterior, for any function $f_1(\cdot)$ we get the following chain-rule $\nabla_{\boldsymbol{\Sigma}_{w_t w_t}} f_1(\cdot) = \nabla_{\boldsymbol{\Sigma}_t} f_1(\cdot) \times \nabla_{\boldsymbol{\Sigma}_{w_t w_t}} \boldsymbol{\Sigma}_t$.

### B.2  Gradient calculation

By using the variational posterior and the chain rule, the gradients of $g$ required for the natural gradient update Eq. (3) is

$$\partial_{\boldsymbol{\mu}^{(2)}} \alpha = \frac{1}{2} \left[ \int_\tau \mathbf{P}_\tau^\top\, \partial_{\boldsymbol{\mu}_\tau\, \boldsymbol{\mu}_\tau}^2\, \alpha_1(\tau) \mathbf{P}_\tau\, \mathrm{d}\tau + \sum_n \mathbf{P}_n^\top\, \partial_{\boldsymbol{\mu}_n\, \boldsymbol{\mu}_n}^2\, \alpha_2(n) \mathbf{P}_n \right], \tag{5}$$

$$\partial_{\boldsymbol{\mu}^{(1)}} \alpha = \int_\tau \mathbf{P}_\tau^\top\, \partial_{\boldsymbol{\mu}_\tau}\, \alpha_1(\tau)\, \mathrm{d}\tau + \sum_n \mathbf{P}_n^\top\, \partial_{\boldsymbol{\mu}_n}\, \alpha_2(n)$$
$$+ \int_\tau \mathbf{P}_\tau^\top\, \partial_{\boldsymbol{\mu}_\tau\, \boldsymbol{\mu}_\tau}^2\, \alpha_1(\tau) \mathbf{P}_\tau\, \boldsymbol{\mu}_{w_\tau}\, \mathrm{d}\tau + \sum_n \mathbf{P}_n^\top\, \partial_{\boldsymbol{\mu}_n\, \boldsymbol{\mu}_n}^2\, \alpha_2(n) \mathbf{P}_n\, \boldsymbol{\mu}_{w_n}, \tag{6}$$

where $\alpha_1(\tau) = \mathbb{E}_{q(\mathbf{x}_\tau)}[h(\mathbf{x}_\tau)]$ and $\alpha_2(n) = \mathbb{E}_{q(\mathbf{x}_n)}[\log p(\mathbf{y}_n \mid \mathbf{x}_n)]$.

## C  Experiment details

The OU SDE parameters used for the simulating the data is $a = -0.5$, $L = 1$, and $q = 0.2$. We simulate the SDE using Euler–Maruyama with the time-step $0.01$ and randomly select $20$ observation samples on it. We use a Gaussian observation model with zero mean and variance of $0.01$. For the psuedo prior, we randomly draw drift and diffusion values from a unit Gaussian. Adam optimizer is used for optimizing the hyperparameters with initial learning rate of $0.1$ and the learning rate for natural gradient descent is set to $0.2$.

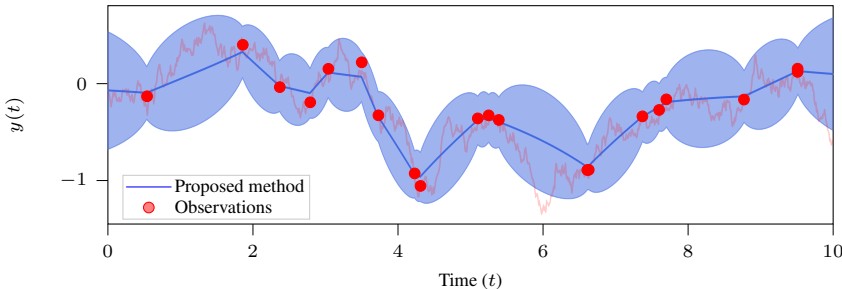

Figure 2: Ornstein–Uhlenbeck process: Approximated posterior mean and 95% confidence interval of the proposed method along with the simulated trajectory and the noisy observations.

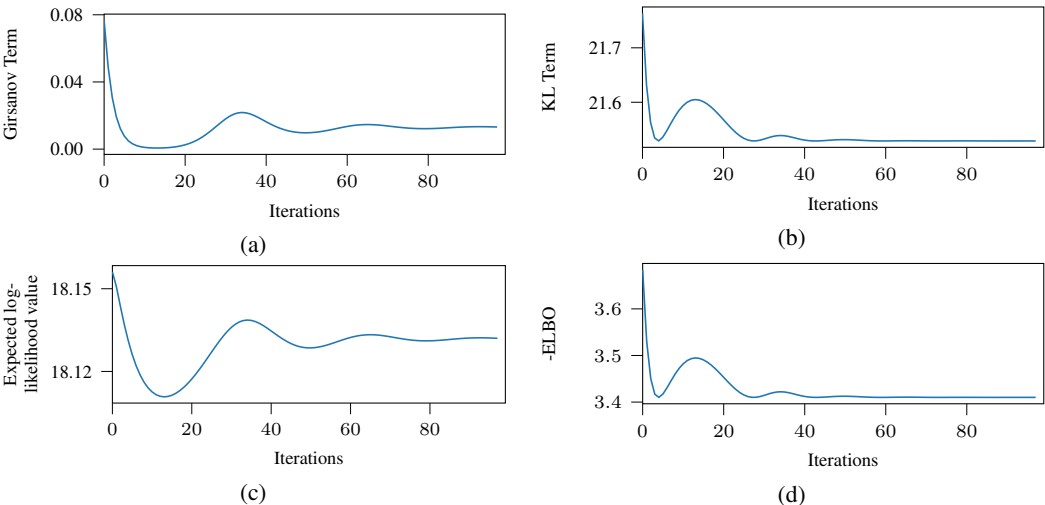

Figure 3: Ornstein–Uhlenbeck process: The evolution of the (a) Girsanov value; (b) Kullbeck–Liebler divergence value; (c) Expected log-likelihood value; (d) Negative ELBO; over training iterations.