# OpenReview forum: "Sparse Gaussian Processes for Stochastic Differential Equations"
_NeurIPS.cc/2021/Workshop/DLDE — DLDE Workshop -- NeurIPS 2021 Poster_

### Official Review · Reviewer_Zvig · 2021-09-28
**Review for Sparse Gaussian Processes for Stochastic Differential Equations**

**Confidence:** 3

**Review:**

# Summary

This paper presents an approach for learning underlying parameters of SDEs, given noisy observed measurements of the state of the system. In particular, they focus on SDEs which include a drift function, constant diffusion coefficient and standard Brownian motion, and attempt to learn parameters of the drift function. To learn the underlying parameters, they use variational inference, fitting an approximate distribution to the posterior of the state, which in turn maximises the marginal likelihood of the observations, whilst jointly updating the parameters of the drift function.

# Main review

To be upfront, I am not an expert in SDEs, but have attempted to give a reasoned review below.

## Originality

This paper is a direct extension of Archambeau et al (2007), the difference being that Archambeau et al use a Markovian GP as the approximate posterior distribution, whereas this paper uses a conditioned Markovian GP. This reduces the complexity of the variational optimisation problem in both memory and time and allows gradient descent algorithms to be used. Thus, this paper appears novel in that it investigates a new family of variational distributions to approximate the posterior distribution.

## Quality

The mathematical description of the method appears well reasoned and sound, and the related work appears adequately cited. However, the experiments appear fairly limited. Specifically, the authors only show results for one particular SDE (Ornstein–Uhlenbeck), where only a single scalar parameter from a linear drift function is learned. The authors do acknowledge that their method performs poorer when using a saw-tooth like drift function; it would be useful to see further results on how the proposed technique handles more complex/ higher dimensional/ other types of SDEs.

The experimental results also lack benchmarks, and so I find it difficult to evaluate how well they perform in a relative sense. It would be useful to compare their results to the method of Archambeau et al, or other simpler baseline approaches. Also, some comparison of the computational/ memory cost compared to Archambeau et al would be useful to support their claim that the proposed method has lower time/memory complexity, and perhaps a discussion on how these costs scale with e.g. number of input dimensions or underlying SDE parameters.

## Clarity

The paper is well structured, and its arguments are logically presented.

## Significance

Without comparison to other state-of-the-art techniques in this area, it is difficult to say how significant this approach is. The key novelty appears incremental (changing the family of variational distributions, rather than proposing an entirely different learning algorithm), although promising.

**Score:**

3: Good paper

---

### Official Review · Reviewer_hDen · 2021-10-10
**Well written paper, but incremental improvement over previous work**

**Confidence:** 3

**Review:**

This work  is a direct extension of Archambeau et al (2007),  where instead of Markovian GP as the approximate posterior distribution, this paper uses a conditioned Markovian GP. It is very well written, with proper citation of the literature and ample explanation of methods and derivations.

However, it still remains an incremental improvement of Archambeau et al (2007), with a linear drift term, and as also acknowledged by the authors, the example of the double well system of Archambeau et al (2007) can not be learned under the current setup. The authors do mention as improvement improvement having a piece-wise stationary  Markovian GP, but one if left to wonder how many inducing points and how densely spread should they be to properly represent a general non-linear drift.
In addition, the example chosen for illustration is rather limited in scope as it is one dimensional linear drift and constant diffusion, with inducing points taken to be the same as the observation points, and no learning step being performed. One glaring omission is the comparison of results to the literature or other approaches. Also, the authors could have commented on the inclusion of various types of colored noise in their setup.
Perhaps, the paper would have benefited by the application and study of improvements mentioned in section 4 before submission.



**Score:**

2: Borderline paper

---

### Official Review · Reviewer_iQ5u · 2021-10-11
**Good writeup of the problem but incremental results**

**Confidence:** 2

**Review:**

### Summary:

This work extends the work  from [1] by introducing a new parametrization of the approximate distribution using a conditioned Markovian Gaussian process. This reparametrization, reduces the computational training cost by allowing parallel evaluations of the marginal posterior prediction. It is also compatible with learning methods such natural gradient descent.

### Review

Over all the authors present a concise and clear summary of the problem of learning SDEs  from observations and in particular learning an Ornstein-Uhlenbecker process. The presentation was well organized and easy to understand even for a novice in SDE's.

As the author's mentioned this is a direct extension from [1] with questionable benefits. Without a direct comparison with other methods it is difficult to judge whether this is an improvement on any axis as claimed. For example, the plot in Figure 1, sparse process drift appears to converge to a constant offset from the True drift. Is this actually good performance when compared to other works or is this a cost for the training speed-up? The proposed method also fails to encompass some measurement observations around t=6. Again, without a quantitative comparison it unclear whether this is expected or a significant error. The authors also mention the computational benefits of their approach but fail to demonstrate this empirically.

Even though their method is well presented and reasonable, a lack of empirical comparisons is the critical weakness of the paper. It would also be interesting to see how framework fits with other approaches to learning SDE's such those presented in [2].

### References
[1] Archambeau, Cedric, et al. "Gaussian process approximations of stochastic differential equations." Gaussian Processes in Practice. PMLR, 2007.

[2] Kidger, Patrick, et al. "Efficient and Accurate Gradients for Neural SDEs." arXiv preprint arXiv:2105.13493 (2021).


### Minor Errors:
Line 77: to obtained


**Score:**

2: Borderline paper

---

### Decision · Program_Chairs · 2021-10-16

**Decision:**

Accept (Poster)

**Comment:**

We want to encourage new researchers to explore new ideas in the DLDE space, so we accept this article.